# Evolution and modulation of antigen-specific T cell responses in melanoma patients

Jani Huuhtanen [1,2,3,4], Liang Chen[5,6], Emmi Jokinen[4], Henna Kasanen[1,2,3], Tapio Lönnberg [7,8], Anna Kreutzman[1,2,3], Katriina Peltola[3,9], Micaela Hernberg[9], Chunlin Wang[5,6], Cassian Yee [10,11], Harri Lähdesmäki[4], Mark M. Davis [5,6,12,14] ✉ & Satu Mustjoki [1,2,3,13,14] ✉

Analyzing antigen-specific T cell responses at scale has been challenging. Here, we analyze three types of T cell receptor (TCR) repertoire data (antigen-specific TCRs, TCR-repertoire, and single-cell RNA + TCRαβ-sequencing data) from 515 patients with primary or metastatic melanoma and compare it to 783 healthy controls. Although melanoma-associated antigen (MAA) -specific TCRs are restricted to individuals, they share sequence similarities that allow us to build classifiers for predicting anti-MAA T cells. The frequency of anti-MAA T cells distinguishes melanoma patients from healthy and predicts metastatic recurrence from primary melanoma. Anti-MAA T cells have stem-like properties and frequent interactions with regulatory T cells and tumor cells via *Galectin9-TIM3* and *PVR-TIGIT* -axes, respectively. In the responding patients, the number of expanded anti-MAA clones are higher after the anti-PD1(+anti-CTLA4) therapy and the exhaustion phenotype is rescued. Our systems immunology approach paves the way for understanding antigen-specific responses in human disorders.

Antigen-specific T cell responses are a hallmark of tumor immunology and fundamental for understanding, detecting, and monitoring the complex tumor-immune cell interactions. Only a small fraction of tumor-infiltrating T cells are specific to cancer antigens while bystanders, i.e., T cells that do not recognize cancer but e.g., viral antigens, are abundant[1–3]. Previous studies have leveraged T cell receptor (TCR) sequencing to understand the diversity and dynamics of the TCR repertoire following immune checkpoint inhibitor therapies in solid cancers[4–7] and have linked crude metrics of TCR repertoire to clinical outcomes. However, these studies have lacked the tools to dissect the

diversity and dynamics of antigen-specific T cells, which could be more important in understanding the role of anti-tumor T cells and bystander T cells in the tumor microenvironment (TME) and could aid in discovering biomarkers for clinically relevant responses. For example, high TCR repertoire clonality has been linked to a favorable outcome to immune checkpoint inhibitor therapy in melanoma[4,6], but in these studies it has not been shown that the clonally expanded T cells are tumor-reactive. Profiling of antigen-specific T cells with panels of peptide and major histocompatibility complex (pMHC) tetramers remains both sample and time consuming[8] and cannot be applied

[1]Translational Immunology Research Program, University of Helsinki, Helsinki, Finland. [2]Hematology Research Unit Helsinki, Helsinki University Hospital Comprehensive Cancer Center, Helsinki, Finland. [3]iCAN Digital Precision Cancer Medicine Flagship, Helsinki, Finland. [4]Department of Computer Science, Aalto University, Espoo, Finland. [5]Department of Immunology and Microbiology, Stanford University, Stanford, CA, USA. [6]Institute for Immunity, Transplantation and Infection, Stanford University School of Medicine, Stanford, CA, USA. [7]Turku Bioscience Centre, University of Turku and Åbo Akademi University, Turku, Finland. [8]InFLAMES Research Flagship Center, University of Turku, Turku, Finland. [9]Department of Oncology, Helsinki University Hospital Comprehensive Cancer Center, Helsinki, Finland. [10]Department of Melanoma Medical Oncology, The University of Texas MD Anderson Cancer Center, Houston, TX, USA. [11]Department of Immunology, The University of Texas MD Anderson Cancer Center, Houston, TX, USA. [12]Howard Hughes Medical Institute, Stanford University School of Medicine, Stanford, CA, USA. [13]Department of Clinical Chemistry and Hematology, University of Helsinki, Helsinki, Finland. [14]These authors jointly supervised this work: Mark M. Davis, Satu Mustjoki. ✉e-mail: mmdavis@stanford.edu; satu.mustjoki@helsinki.fi

retrospectively to previously published next-generation sequencing data. To overcome this, several transcriptional markers have been suggested to be associated with tumor reactivity, like CD39[+](CD103[+]) T cells[2,9,10], but gene expression can serve only as a surrogate marker for tumor-reactivity.

Neoantigens, which have been associated with response to immune checkpoint inhibitor therapies are still imperfect response predictors[11,12]. Additionally, neoantigens are challenging to study as not all somatic mutations cause neoantigens, not all neoantigens are immunogenic, and most importantly, the immunogenic neoantigens are highly restricted to individuals[13]. In contrast, tumor-associated antigens (TAAs) are usually immunogenic, widely expressed by different tumor cells, and shared across individuals[14], which provides a useful way to detect and monitor anti-tumor T cell responses in larger patient cohorts. In long-term melanoma survivors, TAA specific T cells have been shown to form an anti-tumor repertoire that persists after tumor eradication[15] and has an exhausted phenotype, that could perhaps be rescued with immune checkpoint inhibitor therapies[16].

Here, we profile and analyze three types of T cell repertoire data to understand anti-tumor immunity via TAA-specific T cells (Fig. 1a, Supplementary Fig. 1a, and Supplementary Data 1). First, we examine pMHC-tetramer sorted epitope-specific TCRβ-sequencing (TCRβ-seq) data across different melanoma-associated antigens (MAAs) and compare these to similar viral antigen data and use this information to create tools to detect anti-MAA specific T cells. Second, to understand the phenotype of antigen-specific T cells, we analyze previously published scRNA+TCR-seq, paired bulk-RNA, and TCRβ-seq from the melanoma biopsies. Finally, we analyze scRNA+TCR-seq and TCRβ-seq data without known specificity from 515 primary and metastatic melanoma patients to understand the antigen-specific immune responses in a broader cohort of melanoma patients (Fig. 1a). In total, our analysis includes over 1000 samples from 15 different data sets with over 10 million TCRs from patients with melanoma and healthy. With this data set, we show (1) the importance of anti-MAA T cells in separating patients with melanoma from healthy based on a peripheral blood samples, (2) show how anti-MAA T cells are associated with overall survival, (3) how anti-MAA T cells are inhibited by the tumor and other immune populations via druggable axes, (4) that in responders, there are more expanded anti-MAA T cell clones after the immune checkpoint inhibitor therapies and (5) that the exhaustion is reversed in anti-MAA T cells only in responding patients, potentially serving as a therapy response biomarker. In summary, our results provide insights into antigen-specific T cell responses and show how they are modulated by different immunotherapies.

## Results

### Anti-MAA TCRs are patient-restricted, but antigen-specific sequence characteristics are shared

To understand how T cells recognize MAAs, we analyzed TCR repertoires from stage IV melanoma patients ($n = 9$) registered to receive ex vivo expanded MART1/MLANA$_{AAGIGILTV}$-specific T cells in an adoptive cells transfer (ACT) trial[17]. We noted a high degree of variation in the diversity of the MART1$_{AAGIGILTV}$-specific repertoire, and some patients had highly oligoclonal responses and others more polyclonal responses (Fig. 1b). Only a few TCR sequences were shared between patients (6.62%, 96/1450 clonotypes, Fig. 1c), which we confirmed in the reanalysis of similar ACT infusion products of MART1$_{ELAGIGILTV}$ (Supplementary Fig. 1a, b) and MELOE1$_{TLNDECWPA}$-specific T cells[18] (Supplementary Fig. 1c, d). The comparison of two TCR repertoires against two similar epitopes, MART1$_{ELAGIGILTV}$ and MART1$_{AAGIGILTV}$, revealed only a negligible number of shared sequences (0.67%, 18/2680, Supplementary Fig. 1e). But, as noted in twin studies and other work in the literature[19–21], exact TCR sequence matches capture only a small fraction of shared specificities between individuals, since many different sequences can encode TCRs with the same peptide-MHC

specificity, warranting a need for more sophisticated soft-matching methods.

Recent work by us and others have elucidated that antigen-specific signals, like short antigen-interacting amino acid motifs, can be extracted from epitope-specific TCRs with modern machine learning tools[22–25]. Thus, we analyzed the similarity of antigen-specific repertoires by comparing them to naïve repertoires to find the antigen-specific signal with GLIPH 2.0[25]. We built a database by pooling our data together with previously profiled MAA-specific and viral antigen-specific data covering 59,898 TCRβ sequences from 77 epitopes[18,26]. Different levels of convergence to these motifs were observed for distinct antigens, suggesting that not all antigens elicit homogeneous T cell responses across patients (Fig. 1d and Supplementary Fig. 2a–d). Importantly, unlike antigen-specific clonotypes, antigen-specific motifs were shared between patients, showing the convergent nature of antigen recognition (Fig. 1e).

Interestingly, antigen-specific T cells sorted with two MART1-antigen epitopes, MART1$_{AAGIGILTV}$ and MART1$_{ELAGIGILTV}$, showed striking similarity on their TCR repertoires, as their TCRs shared most of their antigen-specific motifs ($P = 5.2e-7$, $R^2 = 0.87$, Spearman's rank correlation, Fig. 1f), implicating that a single TCR could recognize both epitopes. This finding was further validated with crystallography data, as a previously described TCR[27] bound both epitopes with similar conformation and high affinity (Fig. 1g). Furthermore, another TCR[27] containing the second most abundant epitope-specific motif GQP identified by GLIPH 2.0 bound both epitopes with a similar confirmation (Supplementary Fig. 3a).

To systematically analyze the cross-reactivity across epitopes, we calculated the overlap of TCR repertoires of each epitope pair in our data set and validated that especially epitopes that share amino acid level similarities, share TCR motifs (Supplementary Data 2). For example, we found the same motifs as in MART1$_{AAGIGILTV}$ and MART1$_{ELAAGIGILTV}$ TCRs in an analogous epitope arising from a completely different antigen, BST2$_{LLLGIGILV}$, providing evidence that similar epitopes can elicit akin T cell responses even across antigens (BST2$_{LLLGIGILV}$ vs MART1$_{ELAAGIGILTV}$, $P_{adj} = 1.994e-82$, $R^2 = 0.255$, correlation BST2$_{LLLGIGILV}$ vs MART1$_{AAGIGILTV}$ $P_{adj} = 0.003$, $R^2 = 0.064$, Spearman's rank-correlation, Supplementary Fig. 3b–d, Supplementary Data 2). Further, epitopes with amino acid sequence-level similarities that arose from different viral species (DENV1$_{GTSGSPIVNR}$, DENV2$_{GTSGSPIIDK}$, and DENV3/4$_{GTSGSPIINR}$) shared significant numbers of TCR motifs (Supplementary Fig. 4a, b and Supplementary Data 2).

### Anti-MAA TCRs can be predicted with interpretable machine-learning classifiers

After discovering the antigen-specific signal of T cells in detail, we sought to utilize this information to define a machine-learning classifier that could be used as an in silico multimer-sorting strategy for TCR repertoire data where pMHC-tetramer sorting is not feasible. For this purpose, we leveraged TCRGP, our recently described Gaussian process method that estimates the probability that a TCR recognizes a given epitope[23]. We used the same pMHC-sorted TCRβ sequencing data as above and built classifiers for the 5 different MAA epitopes from 4 antigens (2 for MART1 and MELOE1 and two published models for SEC24A and TKT[23]) and compared those to previously published models from endemic viral epitopes from *CMV*, *EBV*, *Influenza A*, and *HSV-2*[23] (training of TCRGP classifiers in "Methods"). For the two MART1-epitopes, whose repertoires showed strong homogeneity, we achieved good prediction accuracies (MART1$_{AAGIGILTV}$ area under the receiver operating characteristic [AUROC] = 0.879, MART1$_{ELAGIGILTV}$ AUROC = 0.739, Fig. 1h, Supplementary Fig. 5a), unlike for MELOE1$_{TLNDECWPA}$ whose repertoire was heterogeneous and did not show convergent motifs (AUROC = 0.617, Supplementary Fig. 5a). Similar observations were found in leave-one-subject-out analyses (Supplementary Fig. 5b). We observed that the prediction accuracy is

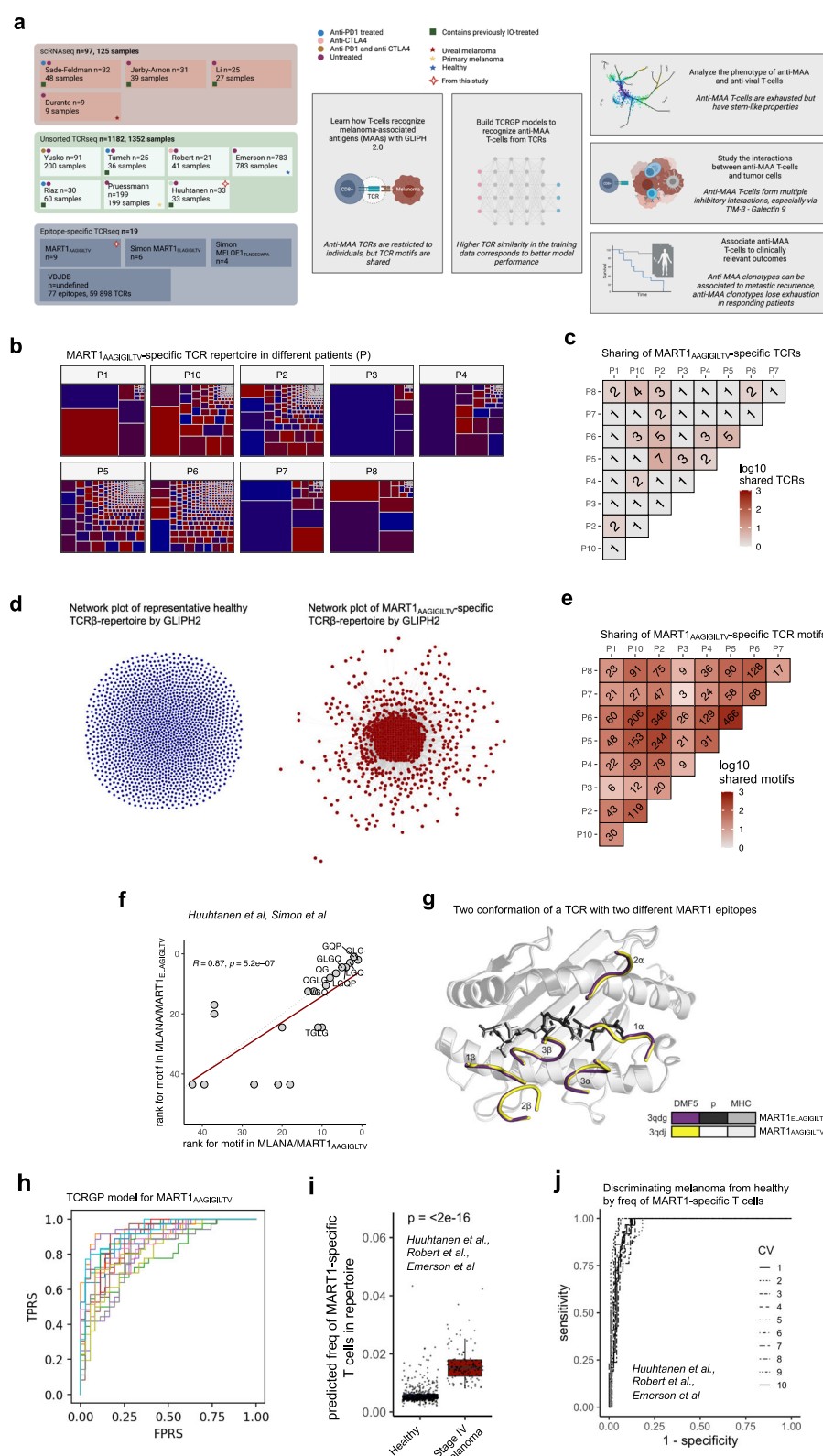

**Nature Communications** | (2022)13:5988

associated with the presence of convergence toward motifs with GLIPH 2.0, providing a biologically meaningful way to interpret our classifiers (Supplementary Fig. 5c).

Further, to better understand false-negatives and false-positives in our classifiers, we calculated precision-recall AUROCs which were comparable or even slightly better than AUROC values, providing confirmation that our TCRGP classifiers perform well when accounting for false-negatives and false-positives (Supplementary Fig. 6a, detailed

analysis in "Methods"). Additionally, the number of clonotypes with a predicted target did not correlate with the amount of inputted number of clonotypes ($P = 0.71$, $R^2 = -0.02$, Spearman's rank correlation, Supplementary Fig. 6b).

Although our epitope-specific TCRβ-seq data used in the training was HLA-restricted, the HLA type is generally not inferable for most publicly available data sets. Therefore, we sought to analyze whether non-HLA-A*02:01 alleles could potentially bind the epitopes where we

**Fig. 1 | MAA-specific TCRs are restricted to individuals, but TCR motifs are shared and provide a signal to learn for soft-matching methods. a** Schematic overview of the study, made with BioRender. **b** Structures of 9 MART1$_{AAGIGILTV}$-specific TCRβ repertoires, where rectangles indicate TCR clonotypes and their frequencies. (Data from this study). **c** The number of shared public clonotypes between the MART1$_{AAGIGILTV}$-repertoires. (Data from this study). **d** Left: The GLIPH 2.0 results of pooled TCRβs from a healthy subject. Each node is a TCRβ clonotype and an edge between nodes denotes a similarity between two TCRβs. Right: Similar GLIPH 2.0 results from pooled MART1$_{AAGIGILTV}$-specific TCRβs from 9 subjects. (Data from this study). **e** The number of shared public TCRβ resolved by GLIPH 2.0 between the MART1$_{AAGIGILTV}$-repertoires. (Data from this study). **f** Rank correlation of GLIPH 2.0 motifs found in two TCR repertoires from similar epitopes, MART1$_{ELAGIGILTV}$, and MART1$_{AAGIGILTV}$, where the ranks are determined by the number of TCRs that share the motif (i.e., motif with the highest rank has the most TCRs). The solid line denotes the linear regression, and the dashed line denotes the diagonal. The correlation coefficient and $P$-value were calculated with two-sided Spearman's rank correlation. (Data from this study and ref. 18). **g** Crystallography schematics of DMF5 TCR showing the positions that can bind both MART1$_{ELAGIGILTV}$ (3qdg) and MART1$_{AAGIGILTV}$ (3qdj) epitopes (p) with identical binding. A similar TCR that was able to bind both epitopes, DMF4, has the QGP motif identified by GLIPH 2.0 (Supplementary Fig. 3a). (Data from ref. 27). **h** ROC curve plot for MART1$_{AAGIGILTV}$ specific TCRβs from leave-one-fold-out analysis when trained with TCRGP. Each line corresponds to one cross-validation from 20-fold cross-validation. (Data from this study). **i** Box plot showing the abundances of predicted MART1-specific clonotypes from healthy donors ($n = 783$) and stage IV melanoma patients ($n = 46$) sampled from peripheral blood. $P$-value was calculated with two-sided Mann–Whitney test. The definition of box plot visualization is stated in the "Methods" section 'Data visualization'. (Data from this study and refs. 5,29). **j** ROC curve plot showing the ROC curve for separating stage IV melanoma patients ($n = 46$) from healthy ($n = 783$) based on information in panel **i**. (Data from this study and refs. 5,29). Source data are provided as a Source Data file.

had training data with a pMHC interaction prediction tool NetMHCpan-4.0[28]. To allow for comparison across HLA molecules, we used the percentile rank scores where lower rank corresponds to a higher probability that a given peptide-MHC interaction naturally appears. We analyzed the predictions for ***GIGILTV, which is the stretch of amino acids shared between the MART1$_{AAAGIGILTV}$ and MART1$_{ELAGIGILTV}$ epitopes, to be presented by different HLA supertypes and found that 6 HLA-supertypes were predicted to bind the 8-10mer epitope with ***GIGILTV within a threshold of <10% rank, implying that epitopes with GIGILTV-motifs could be presented even by non-HLA-A*02:01 carriers (Supplementary Fig. 7a).

## Anti-MAA TCRs can be used to separate patients with melanoma from healthy using blood samples

As the TCRGP classifiers were (1) robust in cross-validation of the epitope-specific data, (2) contain controllable amounts of false-negatives and false-positives, and (3) could possibly be used independently of HLA types, we predicted the abundance of MAA-specific and viral-specific clonotypes from TCR-repertoire level data. For melanoma, we used our unpublished and a published cohort[5] of stage IV melanoma patients with unknown HLA types sampled from peripheral blood before any treatment ($n = 46$), and compared it to the largest known cohort of TCRβ-seq data of 783 healthy donors[29]. Although TCRGP predicted that healthy donors would have anti-MART1 clonotypes, melanoma patients were predicted to have a higher frequency of anti-MART1 clonotypes ($P < 2.2e{-}16$, two-sided Mann–Whitney test, Fig. 1i). By using the frequency of the predicted anti-MART1 clonotypes as an input, we obtained an AUROC of 0.981 for detecting stage IV melanoma patients from healthy in a 10-fold cross-validation with a basic logistic regression model (Fig. 1j). This AUROC was higher than AUROC values calculated using baseline characteristics like patient age or clonality (Supplementary Fig. 8a, b). Importantly, anti-MAA TCRs from these patients were not used as input data to the TCRGP classifiers. Even though there was a difference between the patients with melanoma and healthy groups in age, the amount of anti-MART1 clonotypes or their clonality was not associated with age (Supplementary Fig. 8c). Reassuringly, when we performed subsampling to our cohort to gain an age-matched cohort (stage IV melanoma $n = 25$, healthy $n = 50$), the frequency of anti-MART1 clonotypes was significantly higher in melanoma than in healthy ($P = 1.7e{-}10$, two-sided Mann–Whitney) and with a high AUROC for detecting stage IV melanoma from healthy (AUROC = 0.950, Supplementary Fig. 8d, e).

To prove that our analysis was not separating only HLA biases, we tested other non-melanoma antigens from *CMV*, *EBV*, and *Influenza A*. For *CMV* and *Influenza A*, we did not find any difference between patients with melanoma and healthy, but interestingly patients with melanoma had more TCRs predicted to be reactive against two out of three *EBV* antigens, BMLF1 ($P = 5.2e{-}5$, two-sided Mann–Whitney test) and BRLF1 ($P = 2.3e{-}13$, Supplementary Fig. 8f), which is of interest as

an expansion of a cross-reactive TCR-specificity group against *EBV* and non-small cell lung carcinoma has been reported previously[21]. Hence, we argue that TCRGP can enrich anti-MAA clonotypes and can help to uncover meaningful insights into tumor responses.

## Anti-MAA clonotypes include both stem-like and exhausted T cells, which interact frequently in the tumor microenvironment

Next, we wanted to study the phenotypes of antigen-specific clonotypes and predicted the antigen-specificities of tumor-infiltrating lymphocytes (TILs) from melanoma biopsies profiled with scRNA+TCRβ-seq[30] ($n = 25$, Fig. 2a). TCRGP predicted a higher number of TILs to be reactive against any of the 5 MAA epitopes in comparison to viral epitopes with a false positive rate (FPR) of 5% ($P = 0.029$, one-sided Fisher's exact test). This also included the most expanded clonotype in the data set (80 cells), which was found both in the primary tumor and in a metastatic site from the same patient (Supplementary Fig. 9a).

Importantly, the anti-MAA clonotypes were enriched in the TCF7+ T cell cluster with stem-like properties ($P = 2.2e{-}16$, one-sided Fisher's exact test), whose abundance has been associated with a response to immune checkpoint inhibitor therapies[31], and in a cycling, exhausted CD8+ T cell phenotype ($P = 0.0001$), associated with tumor reactivity in the original publication[30] (Fig. 2b, c). In a cluster agnostic analysis, we calculated the differentially expressed (DE) genes between all anti-melanoma and anti-viral T cells to overcome the possible biases and type I errors made in the clustering. The most differentially upregulated genes in anti-melanoma clonotypes included genes associated with activated, cytotoxic functions in CD8+ T cells (e.g., *NKG7*, *GZMA*, *GZMK*), different cytokines (*CCL4*, *CCL5*), T cell activation (*CD74*, *HLA-DRA*), and inhibitory markers such as *HAVCR2* (*TIM3*), arguing further for their role as tumor-reactive cells (Fig. 2d and Supplementary Data 3). In contrast, the most upregulated genes in cells predicted to be anti-viral included genes related to naïve T cell properties, such as *IL7R* and *SELL* (*L-selectin*). In comparison to anti-viral clonotypes, anti-MAA clonotypes upregulated IFN-γ and IFN-α response pathways ($P_{adj} = 0.023$ and $P_{adj} = 0.010$, GSEA test, Supplementary Fig. 9b), which have been previously reported as biomarkers for an effective response to immune checkpoint inhibitor therapy[32].

Next, we studied the interactions between antigen-specific clonotypes, tumor cells, and other immune subsets by calculating the significant ligand-receptor interaction pairs with a permutation test implemented in CellPhoneDB[33]. In comparison to the anti-viral T cells, the anti-MAA T cells had over two-fold more interactions with B-cells and regulatory T cells (T$_{regs}$) and interacted more frequently with the two tumor clusters than the anti-viral T cells (Fig. 2e and Supplementary Data 4). To address the nature of these interactions, we classified the ligand-receptor pairs and noted several inhibitory interactions. These inhibitory interactions included *PVR* (CD155) expressed by the tumor cells, but not by the other immune cells, and its receptors *TIGIT*

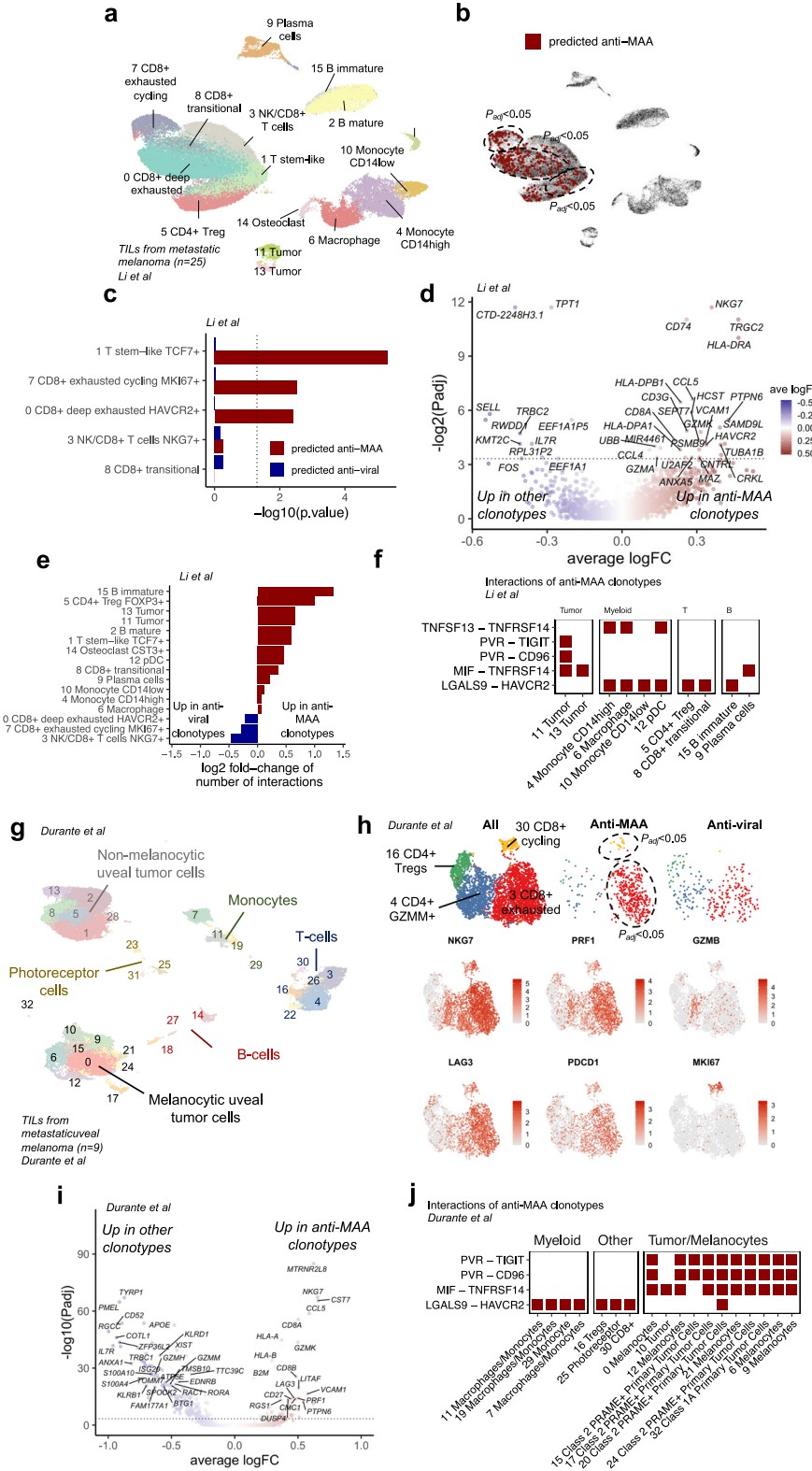

and *CD96* expressed by the anti-MAA T cells and not by the bystander cells, rendering anti-TIGIT therapy as an interesting target to invigorate anti-MAA T cells and not bystanders (Fig. 2f). Conversely, *LGALS9* (encoding *Galectin 9*) was expressed by other immune cells including T$_{regs}$ and myeloid cells and its receptor *HAVCR2* (encoding *TIM3*) was expressed by the anti-MAA T cells, making anti-TIM3 therapy an attractive target in resolving inhibitory interactions with other immune cells (Fig. 2f).

To validate and compare our findings in cutaneous melanoma we reanalyzed a scRNA+TCRαβ-seq dataset including immune and tumor cells from patients with uveal melanoma (*n* = 9)[34], which has a lower mutation burden and immune checkpoint inhibitor therapy response rate than cutaneous melanoma[35] but expresses also same MAAs like MART1[36] and has currently an approved anti-MAA TCR based therapy[37]. As in cutaneous melanoma, the anti-MAA clonotypes were enriched in uveal melanoma in the exhausted CD8+ T cell cluster (*P* < 2.2e−16,

**Fig. 2 | Anti-MAA T cells are exhausted but retain stem-like properties and form the most inhibitory interactions with T$_{reg}$ and tumor cells. a** UMAP projection of expression profiles of tumor-infiltrating lymphocytes (TIL) from 25 treatment naïve or immunotherapy resistant metastatic melanoma patients, colored by cluster. (Data from ref. 30). **b** The same UMAP projection as in panel A where the TCRGP-predicted anti-MAA and anti-viral T cells are highlighted. Melanoma-specific clonotypes include T cells that were predicted by their TCRs to be reactive against epitopes from melanoma-associated antigens MART1$_{AAGIGILTV}$, MART1$_{ELAGIGILTV}$, MELOE1$_{TLNDECWPA}$, TKT$_{AMFWSVPTV}$, and SEC24A$_{FLYNLLTRV}$. Encircled lines denote enrichment ($P_{adj} < 0.05$, Benjamini−Hochberg corrected one-sided Fisher's exact test) to the clusters shown in panel A. (Data from ref. 30). **c** Bar plot showing the enrichment of anti-MAA and anti-viral clonotypes to different T cell phenotypes as $P$-values from Fisher's one-sided exact test. (Data from ref. 30). **d** The differentially expressed genes ($P_{adj} < 0.05$, Bonferroni corrected two-sided t-test) between anti-MAA and anti-viral clonotypes in patients with melanoma. (Data from ref. 30). **e** Bar plot showing the log2 fold-change of the number of statistically significant ($P < 0.05$, two-sided permutation test from CellPhoneDB) ligand−receptor interactions between anti-MAA or anti-viral clonotypes and the cell types found in the

tumor microenvironment as predicted by CellPhoneDB permutation test. (Data from ref. 30). **f** The statistically significant ($P < 0.05$, two-sided permutation test from CellPhoneDB) inhibitory ligand-receptor pairs between anti-MAA clonotypes and cells in the tumor microenvironment of melanoma. (Data from ref. 30). **g** UMAP projection of expression profiles of TILs from patients with primary or metastatic uveal melanoma ($n = 9$), colored by cluster. (Data from ref. 34). **h** Focused UMAP of T cells with recovered TCRs from panel (**g**). On the top row, the clustering is colored by T cell phenotypes and the TCRGP-predicted anti-MAA and anti-viral clonotypes are shown. The dashed circles denote enrichment ($P_{adj} < 0.05$, Benjamini−Hochberg corrected one-sided Fisher's exact test) to clusters shown in panel **h**. A selection of the expression of canonical markers used to define the clusters are highlighted. (Data from ref. 34). **i** The differentially expressed genes ($P_{adj} < 0.05$, Bonferroni corrected two-sided t-test) between anti-MAA and anti-viral clonotypes in patients with uveal melanoma. (Data from ref. 34). **j** The statistically significant ($P < 0.05$, two-sided permutation test from CellPhoneDB) inhibitory ligand-receptor pairs between anti-MAA clonotypes and cells in the tumor micro-environment of uveal melanoma (Data from ref. 34). Source data are provided as a Source Data file.

---

one-sided Fisher's exact test, Fig. 2g, h), a cluster that upregulated *PDCD1*, *HAVCR2 (TIM3)*, and *LAG3*. Also, in a cluster-agnostic analysis, the anti-MAA clonotypes overexpressed cytotoxicity and exhaustion-related genes e.g., *NKG7*, *CCL5*, *PRF1*, and *LAG3* (Fig. 2i and Supplementary Data 3), showing convergence across tumor types. As in cutaneous melanoma, the anti-MAA clonotypes formed frequent inhibitory interactions with tumor cells via *PVR − TIGIT/CD96* and with immune cells via *Galectin 9 − TIM3* (Fig. 2j and Supplementary Data 4).

### Anti-MAA clonotypes are associated with survival in primary melanoma

After revealing the exhausted phenotype of MAA-specific T cells, we examined how these clonotypes are associated with the tumor stage. We analyzed a recently published dataset including TCRβ-seq repertoires from 199 primary melanomas that were completely resected and had not produced metastases at the time of diagnosis[38]. Surprisingly, in a multivariable analysis the absence of a dominating anti-MART1 clonotype (explaining at least >1% of the total repertoire) was associated with later metastatic recurrence and better overall survival in these patients ($P = 0.021$, hazard-ratio [HR] = 0.54), unlike the absence of a dominating clone with any other epitope specificity ($P = 0.431$, HR = 0.84, Fig. 3a, "Methods"). The $P$-value for absence of a dominating anti-MART1 clonotype was smaller than with age ($P = 0.039$, HR = 1.01) or mitotic range ($P = 0.047$, HR = 1.02) and comparable to that with Breslow class ($P = 0.018$, HR = 1.13).

### Anti-MAA clonotypes expand following immune checkpoint inhibitor therapies especially in the responders

To detect whether the pre-existing anti-MAA clonotypes in the tumor expand following immune checkpoint inhibitor therapy or new ones are recruited from the circulation, we gathered TCRβ-seq data from 213 longitudinal samples from 3 cohorts of melanoma patients, containing 100 melanoma patients treated with anti-PD1 with or without anti-CTLA4[4,6,7]. The proportion of MAA-specific T cells was between 5 and 10% of the total repertoire in the tumor, which was higher than in blood samples ($P = 0.00015$, two-sided Mann−Whitney test, Supplementary Fig. 10a).

We calculated the expanded clonotypes between pre and post therapy samples with a Fisher's two-sided exact test ("Methods"), and noticed that patients with a response (complete response [CR] or partial response [PR] as defined by the RECIST criteria[39]) had significantly more expanded clonotypes ($P_{adj} < 0.05$, Benjamini−Hochberg corrected two-sided Fisher's exact test) than patients without a response (stable disease [SD] or progressive disease [PD]) ($P = 0.031$, two-sided Mann−Whitney test, Fig. 3b and Supplementary Data 6). When analyzing the specificities of the expanded clones, we noticed that the most of the clonotypes did not have predicted target as

expected (Fig. 3c). However, out of the clones with a predicted target, anti-MAA clonotypes were more common than anti-viral clonotypes (Fig. 3c). Furthermore, anti-MAA clonotypes expanded more frequently in patients with a response than without ($P = 0.041$), but this was not noted with anti-viral clones (Fig. 3c).

### Clonal replacement is not associated with response to immune checkpoint inhibitor therapy in melanoma

Clonal replacement of TIL TCRs following immune checkpoint inhibitor therapy has recently been analyzed in non-melanoma skin cancers, non-small cell lung cancer, endometrial cancer, colorectal carcinoma, and renal cell carcinoma[40–43]. However, clonal replacement has not been studied in melanoma, and its association with response to immune checkpoint inhibitor therapies is under debate. In our extensive cohort, the frequency of replacing clones (i.e., clones that were not found in pre-treatment samples) was higher in patients who received immune checkpoint inhibitor therapy as a first-line treatment (IO naïve) than in patients previously treated unsuccessfully with immunotherapy ($P = 0.031$, two-sided Mann−Whitney test, Fig. 3d). Unlike in a previous report on non-melanoma skin cancer[40], high frequency of clonal replacement after immune checkpoint inhibitor therapy was associated with a worse response in IO naïve patients ($P = 0.071$), whereas the frequency of replacing clones was the same in previously immunotherapy treated patients ($P = 0.97$, Fig. 3e). Neither the number nor clonality of the replacing clones was associated with a response (Supplementary Fig. 11a−c). When the epitope specificities of the replacing clones were predicted with TCRGP, we found that the most clones were specific to MAAs, and the rate of replacement of anti-MAA clones was higher in the non-responding patients than in the responding patients ($P = 0.053$, Fig. 3f).

### Immune checkpoint inhibitor therapies reverse the exhaustion of anti-MAA clonotypes in responders

As anti-MAA clonotypes did not significantly expand following immune checkpoint blockade, we wanted to understand whether immune checkpoint inhibitor therapies induce phenotypic alterations in anti-MAA T cells. We predicted the antigen-specificities of published scRNA+TCRαβ-seq dataset including TILs from both prior and post-anti-PD-1 alone or with anti-CTLA-4 treated patients[31] ($n = 32$, Fig. 4a). First, we were able to validate our finding that anti-melanoma clonotypes are enriched to an exhausted CD8+ cluster ($P = 0.008$, one-sided Fisher's exact test, Fig. 4a, b). Second, we noted that anti-MAA T cells lost their exhaustion in patients with complete or partial responses ($n = 18$, $P = 0.0068$, two-sided Mann−Whitney test), but not in non-responders ($n = 14$, $P = 0.92$, Fig. 4c). Notably, the exhaustion phenotype of anti-viral T cells was not affected by the immune checkpoint inhibitor treatment in either response group. The T cells without

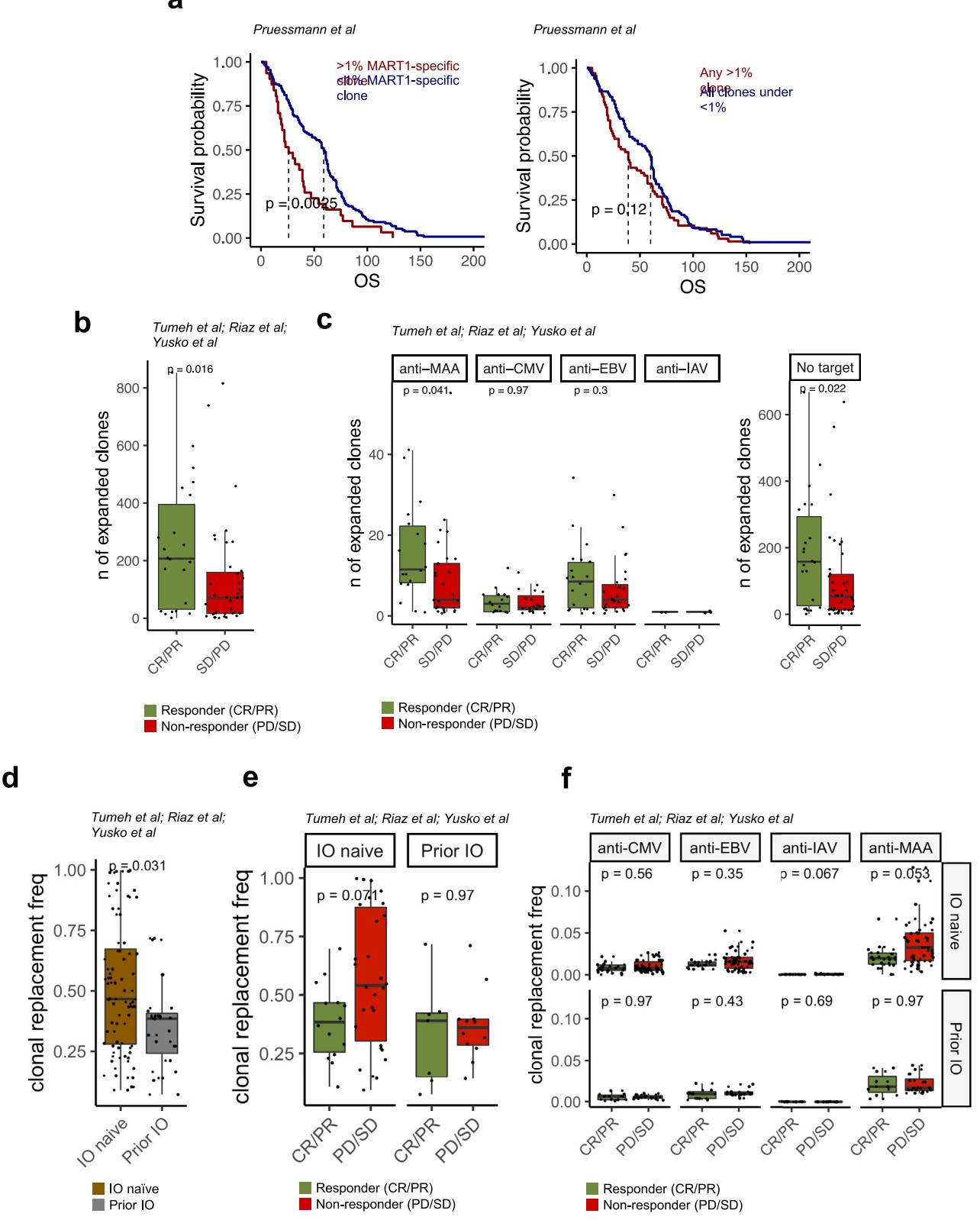

predicted epitope-specificity, possibly containing a mix of other tumor-reactive and bystander T cells, also lost exhaustion in the responding patients but not in non-responding patients ($P = 0.0016$, Supplementary Fig. 12a).

## Discussion

As multiple different TCR repertoire studies have been reported in patients with melanoma, we were able to perform a pooled analysis of

the antigen-specific T cell responses. Although numerous different TCR repertoire algorithms that group TCRs based on epitope specificities or predict epitope binding, have been proposed[44–46], only a few studies have utilized these to analyze large TCR-seq or scRNA+TCR-seq cohort(s)[21]. These studies have usually relied on unsupervised methods, used data from one institution or batch, and, most importantly, used data profiled only with one omics-method. With our in-depth analysis of three different types of T cell repertoire data in primary and metastatic

**Fig. 3 | The number of expanded anti-MAA clonotypes increases following immune checkpoint inhibitor therapy. a** Kaplan–Meier curves of overall survival (OS) for primary melanoma patients ($n = 199$) by the presence of a dominant clonotype (explaining >1% of the total repertoire) that was predicted to recognize MART1 or by any clone explaining >1% of the total repertoire. *P*-values were calculated with two-sided log-rank test. (Data from ref. [38]). **b** Box plot showing the number of expanded clonotypes in the tumor microenvironment following anti-PD1 with or without anti-CTLA4 therapy, where each dot is one individual patient ($n = 60$). Patients were divided by the response (responders $n = 22$, including complete response [CR] and partial response [PR], non-responders $n = 38$, including stable disease [SD], and progressive disease [PD], defined by RECIST criteria). *P*-value was calculated with two-sided Mann–Whitney test. (Data from refs. 4,6,7). **c** Similar box plot as in panel B but where the clonotypes are separated by their specificity. Melanoma-specific clonotypes include T cells that were predicted by their TCRβ CDR3 parts to be reactive against epitopes from melanoma-associated antigens MART1$_{AAGIGILTV}$, MART1$_{ELAGIGILTV}$, MELOE1$_{TLNDECWPA}$, TKT$_{AMFWSVPTV}$, and SEC24A$_{FLYNLLTRV}$. Similarly, the viral-specific antigens include CMV pp65$_{IPSINVHHY}$,

CMV pp65$_{NLVPMVATV}$, CMV pp65$_{TPRVTGGGAM}$, EBV BMLF1$_{GLCTLVAML}$, EBV BRLF1$_{YVLDHLIVV}$ EBV BZLF1$_{RAKFKQLL}$ and IAV M1$_{GILGFVFTL}$. *P*-values were calculated with two-sided Mann–Whitney test. (Data from refs. 4,6,7). **d** The abundance of replacing clones after immune checkpoint therapy, i.e., clones that are found in the tumor microenvironment only following anti-PD1 with or without anti-CTLA4 therapy. The patients ($n = 60$) are divided by the previous line of immune checkpoint therapy (IO naïve $n = 46$, prior IO $n = 15$). *P*-values were calculated with two-sided Mann–Whitney test. (Data from refs. 4,6,7). **e** The abundance of replacing clones after checkpoint therapy divided by prior line of immunotherapy (IO naïve CR/PR $n = 18$, SD/PD $n = 28$; prior IO CR/PR $n = 4$, SD/PD $n = 10$). *P*-values were calculated with two-sided Mann–Whitney test. (Data from refs. 4,6,7). **f** The abundance of clonal replacement clones after checkpoint therapy divided by predicted specificity as in panel (**e**). *P*-values were calculated with two-sided Mann–Whitney test. (Data from refs. 4,6,7). The definition of box plot visualization is stated in the "Methods" section 'Data visualization'. Source data are provided as a Source Data file.

cutaneous melanoma, we were able to discover similarities in the T cell recognition of melanoma epitopes and build models to predict melanoma antigen-specific T cells providing in-depth understanding of their frequency, phenotype, and association with prognosis and therapy response in a large cohort of melanoma patients (Fig. 4d).

With our machine learning models, we were able to separate patients with melanoma from healthy only based on the proportion of anti-MART1 clones in a blood sample. Importantly, this method could provide an avenue for early detection of cancer if this finding can be replicated in patients with low grade melanoma. Moreover, we discovered that the presence of a dominating anti-MART1 clone (>1% repertoire) in the tumor is associated with a worse survival in primary melanoma. We speculate that the large MART1-clones could be more prone to exhaustion and thus cannot mount an adequate long-standing anti-tumor immune response. Further clinical studies are needed to understand, whether immune checkpoint inhibitor therapies could reverse this worse survival.

Further, we were able to link the predicted target of the T cells to their phenotype and show the differential phenotypes of anti-MAA and anti-viral T cells. Anti-MAA-clonotypes showed expression of genes that have been previously associated with tumor-specificity and cytotoxicity (e.g., *NKG7, CCL5, TIM3*), as in a previous analysis of anti-MAA and neoantigen-specific clones in a scRNA-seq data set[16]. The retained proliferation capacity of the anti-MAA clonotypes at the transcriptome level implies that not only neoantigen-specific T cells can be attributed to benefit from the immune checkpoint blockade, as MAA-specific T cells could also play a role. Additionally, we provide a systems immunology overview of immune subset interactions between T cells of different specificities and suggest that the exhaustion of anti-MAA T cells could be caused by interactions with Tregs, especially via *Galectin9* and *TIM3*, and with tumor cells via *PVR* and *TIGIT*, posing a potentially interesting strategy to invigorate anti-MAA and tumor cell interactions with anti-TIGIT and anti-MAA and immune cell interactions with anti-TIM3 antibodies, which are both currently tested in phase III trials in various solid and hematological cancers[47,48].

The patients responding to immune checkpoint inhibitor therapy had higher number of expanded clonotypes after the treatment than patients without a response, and many of these clonotypes were predicted to be specific to MAAs. The expansion of anti-MAA clonotypes was congruent with the scRNA-seq data, in which also exhaustion of MAA-specific T cells was successfully reversed with anti-PD1(+anti-CTLA4) therapies. Our work validates and extends the experimental data coming from a limited number of samples, suggesting that MAA-specific T cells selectively lose their exhaustion during immune checkpoint inhibitor therapies[16,49].

The question of clonal replacement, i.e., whether the immune checkpoint inhibitor therapies recruit new clones or expand the

pre-existing ones, is under debate[6,30,40,41,43]. In our analysis of multiple cohorts of melanoma patients, we were not able to associate clonal replacement of the total repertoire or anti-MAA repertoire to clinical responses. However, clonal replacement was more abundant in immune checkpoint therapy naïve patients than in patients who have failed previous line(s) of immunotherapy. Similarly, it was more prevalent in anti-MAA T-cells than in bystander T cells. Altogether, we believe that pre-existing anti-tumor immunity, anti-MAA, and/or anti-neoantigen immunity, in the tumor may be the most important thing to successful immune checkpoint therapy.

Our study has limitations such as the lack of HLA information from the samples, the limited set of anti-MAA epitopes available to train the TCRGP classifiers, and the lack of training data against neoantigen epitopes. As the amount of epitope specific TCR data increases and shared neoantigens and their cognate TCRs are recognized, these prediction tools become even more accurate which could allow their use in the clinical diagnostics.

In summary, our analysis of a large data set of antigen-specific T cells in patients with melanoma gives us insights into the T cell responses, including reversal of exhaustion, expansion of anti-MAA clonotypes, downregulation of target antigens, and molecular mimicry. These findings also provide insights into how different immune checkpoint therapies modulate the response. We envision similar approaches will provide more detailed information of antigen-specific responses at a larger scale in tumor-, auto-, and alloimmunity when suitable training data for TCRs specific for self- or viral antigens emerge.

## Methods

The study was conducted in accordance with the Declaration of Helsinki complying with all relevant ethical regulations. Written informed consent was received from all patients. No compensation was provided for the study participants.

### Metastatic melanoma patients

**MART1$_{AAGIGILTV}$-specific adoptive cell therapy infusion products (Huuhtanen et al. data).** This study included the adoptive cell therapy (ACT) infusion products of 9 patients with metastatic melanoma enrolled to receive such therapy. The study was approved by the Fred Hutchinson Cancer Research Center Institutional Review Board and all patients provided written informed consents. Briefly, the therapy included infusion of autologous MART1$_{AAGIGILTV}$-specific cytotoxic lymphocytes generated by priming with peptide-pulsed dendritic cells in the presence of interleukin-21 and enriched by peptide-major histocompatibility complex multimer-guided cell sorting CTLs followed by a standard course of anti-CTLA4. The patient details are outlined in the original publication[17]. The patient data, including the response

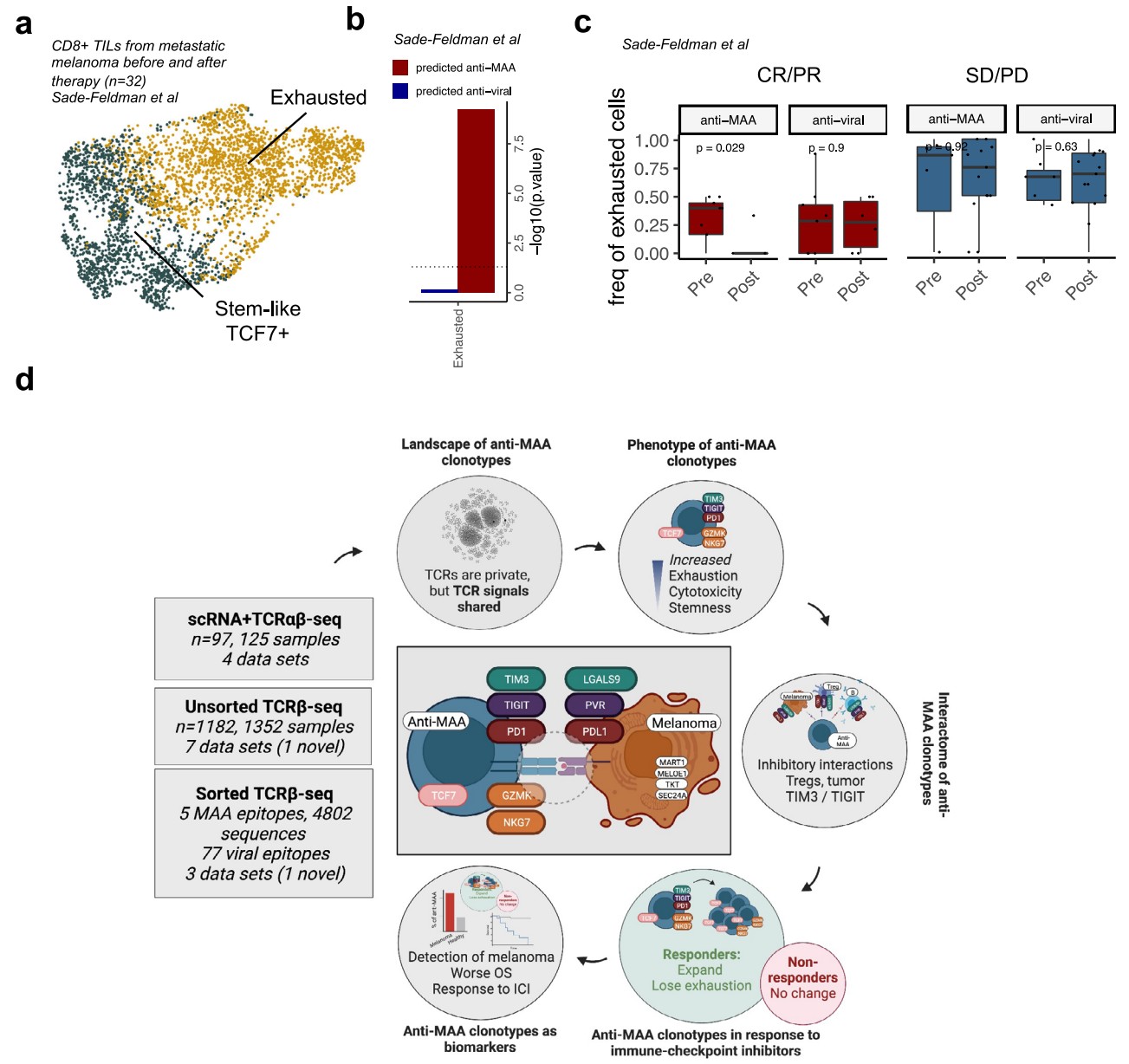

**Fig. 4 | The exhausted phenotype of anti-MAA T cells is reinvigorated with immune checkpoint-therapies in responding patients, unlike in anti-viral T cells. a** UMAP projection of expression profiles of CD8[+] tumor-infiltrating lymphocytes with identified TCR from 48 metastatic melanoma samples from 32 patients before and after anti-PD1 with or without anti-CTLA4, therapy, colored by cluster. (Data from ref. 31). **b** Bar plot showing the enrichment of anti-MAA and anti-viral clonotypes to exhausted T cell phenotype as *P*-values from one-sided Fisher's exact test. (Data from ref. 31). **c** Boxplot showing the frequency of exhausted cells before and after immune checkpoint therapy. Patients were divided by the response (responders *n* = 18, including complete response [CR] and partial response [PR], non-responders *n* = 14, including stable disease [SD], and progressive disease [PD], defined by RECIST criteria). *P*-values were calculated with

paired Mann–Whitney test. T cells were divided by TCRGP predicted specificities, where melanoma-specific clonotypes include T cells that were predicted by their TCRβ to be reactive against epitopes from melanoma-associated antigens MART1$_{AAGIGILTV}$, MART1$_{ELAGIGILTV}$, MELOE1$_{TLNDECWPA}$, TKT$_{AMFWSVPTV}$, and SEC24A$_{FLYNLLTRV}$. Similarly, the viral-specific antigens include CMV pp65$_{IPSINVHHY}$, CMV pp65$_{NLVPMVATV}$, CMV pp65$_{TPRVTGGGAM}$, EBV BMLF1$_{GLCTLVAML}$, EBV BRLF1$_{YVLDHLIVV}$, EBV BZLF1$_{RAKFKQLL}$, and InfA M1$_{GILGFVFTL}$. The definition of box plot visualization is stated in the "Methods" section 'Data visualization'. (Data from ref. 31). **d** The figure summarizes the used datasets, research questions addressed, and the main findings of the study. The figure was made with BioRender. Source data are provided as a Source Data file.

status based on the RECIST criteria[39], were gathered from the respective articles.

**Metastatic melanoma peripheral blood samples (Huuhtanen et al. data).** This study included 33 metastatic melanoma patients who were treated at the Helsinki University Hospital Comprehensive Cancer Center, Finland (Helsinki cohort). Out of the total 33 patients, 9 were immune-oncology (IO) naïve and 24 patients had been previously treated with either anti-PD1 monotherapy or in combination

with anti-CTLA4. The study was approved by Helsinki University Central Hospital (HUCH) ethical committee (Dnro 115/13/03/02/15). Written informed consent was received from all patients and the study was conducted in accordance with the Declaration of Helsinki. Peripheral blood (PB) samples were obtained from the patients before the initiation of IO treatment (IO naïve cohort) or before the start of new treatment modality (prior IO treated cohort). Peripheral blood mononuclear cells (PB MNCs) were separated using the Ficoll-Paque density gradient centrifugation (GE Healthcare, Buckingham,

UK) and were live frozen at −150 °C in 10% DMSO-FBS solution for further assays. The patient details are outlined in Supplementary Data 1.

### TCRβ-sequencing data profiling and acquisition

**MART1$_{AAGIGILTV}$-specific adoptive cell therapy infusion products (Huuhtanen et al. data).** TCRβ-sequencing was conducted as previously described[50,51] with ImmunoSEQ assay by Adaptive Biotechnologies Corp. Genomic DNA was used in all cases.

**MART1$_{ELAGIGILTV}$ and MELOE1$_{TLNDECWPA}$-specific adoptive cell therapy infusion products (Simon et al. data).** TCRα and TCRβ-seq data specific for MART1$_{ELAGIGILTV}$ and MELOE1$_{TLNDECWPA}$ epitopes were gathered from Simon et al.[13], where PBMCs from HLA-A*02+ patients were stimulated with IL-2 and given peptide in a microculture, and the RNA was sequenced with a UMI based technique.

**Metastatic melanoma peripheral blood samples (Huuhtanen et al. data).** 33 patients with metastatic melanoma provided 33 samples that were profiled with TCRβ-seq from MNC from peripheral blood (Helsinki cohort). TCRβ-sequencing was conducted as previously described[50,51] with ImmunoSEQ assay by Adaptive Biotechnologies Corp. Genomic DNA was used in all cases.

**Other epitope-specific data.** Additionally, we gathered data set from VDJdb[26] which is the largest database that contains TCR sequences with known antigen specificity. Entries in VDJdb have been given a confidence score between 0 and 3. To build more reliable classification models with TCRGP[23], we constructed data set so that we selected all epitopes that have at least 50 TCRβ sequences with a confidence score of at least 1 and found data for 12 such epitopes, including MAA TKT$_{AMFWSVPTV}$, MAA SEC24A$_{FLYNLLTRV}$, MAA MART1$_{ELAGIGILTV}$, Influenza A M1$_{GILGFVFTL}$, EBV BMLF1$_{GLCTLVAML}$, CMV pp65$_{IPSINVHHY}$, CMV pp65$_{NLVPMVATV}$, EBV BZLF1$_{RAKFKQLL}$, HSV2 B7$_{RPRGEVRFL}$, CMV pp65$_{TPRVTGGGAM}$, EBV BRLF1$_{YVLDHLIVV}$. For the training and testing of the models, we also required some background TCRs that we do not expect to recognize the epitopes in our data sets, which were gathered from background TCRs constructed in the previous publication[23].

**Metastatic melanoma biopsies before and after anti-PD1 therapy (Tumeh et al. data).** From the Tumeh et al. data[4], 25 patients with metastatic melanoma receiving anti-PD1 therapy provided 36 prior and post-therapy samples that were profiled with TCRβ-seq from MNC sorted from tumor biopsies, and the data was acquired from immuneAccess: https://clients.adaptivebiotech.com/pub/weber-2018-cir.

**Metastatic melanoma samples before and after anti-CTLA4 therapy (Robert et al. data).** From the Robert et al. data[5], 21 patients with metastatic melanoma receiving anti-CTLA4 therapy provided 41 prior and post-therapy samples that were profiled with TCRβ-seq from MNC sorted from peripheral blood, and the data was acquired from immuneAccess: https://clients.adaptivebiotech.com/pub/robert-2014-CCR.

**Metastatic melanoma biopsies and peripheral blood samples before and after anti-PD1 + anti-CTLA4 therapy (Riaz et al. data, Yusko et al. data).** From the Yusko et al. data[7], 91 patients with metastatic melanoma receiving either anti-PD1+anti-CTLA4 therapy or anti-CTLA4+anti-PD1 therapy as a frontline therapy provided 200 prior and post-therapy samples that were profiled with TCRβ-seq with either MNC sorted from tumor biopsies or CD8+ sorted from peripheral blood, and the data was acquired from immuneAccess: https://clients.adaptivebiotech.com/pub/weber-2018-cir.

From the Riaz et al. data[6], 30 patients with metastatic melanoma receiving anti-PD1 either as a front-line or after a previous line of immunotherapy provided 60 prior and post-therapy samples that were profiled with TCRb-seq from MNC sorted tumor biopsies, and the data was acquired from GitHub: https://github.com/riazn/bms038_analysis.

**Primary melanoma patients (Pruessmann et al. data).** From the Pruessmann et al. data[38], 199 patients with primary melanoma provided 199 primary melanoma samples that were profiled with TCRβ-seq from MNC sorted tumor biopsies, and the data was acquired from immuneAccess: https://clients.adaptivebiotech.com/pub/pruessmann-2019-nc.

### scRNA+TCR-seq and bulk-RNA-seq data profiling and acquisition

**Metastatic melanoma biopsies from treatment naive or immunotherapy resistant patients (Li et al. data, Jerby-Arnon et al. data).** From the Li et al. data[30], 25 patients with metastatic melanoma who were treatment naïve or immunotherapy resistant provided 27 samples that were profiled with scRNA+TCRβ-seq from CD45+ enriched tumor biopsies, and the data was acquired from Gene Expression Omnibus (GEO) with accession number GEO: GSE123139.

From the Jerby-Arnon et al. data[52], 31 patients with metastatic melanoma who were treatment naïve or immunotherapy resistant provided 39 samples that were profiled with scRNA-seq with from CD45+ enriched and depleted tumor biopsies, and the data was acquired from GEO with accession number GEO: GSE115973.

**Uveal metastatic melanoma biopsies from untreated patients (Durante et al. data).** From the Durante et al. data[34], 9 patients with uveal melanoma who were treatment naïve provided 9 samples that were profiled with scRNA+TCRαβ-seq with from unsorted tumor biopsies, and the data was acquired from GEO with accession number GEO: GSE115973.

**Metastatic melanoma biopsies before and after immune checkpoint blockade (anti-PD1, anti-PD1 + anti-CTLA4) (Sade-Feldman et al. data, Riaz et al. data).** From the Sade-Feldman et al. data[31], 32 patients with metastatic melanoma who were treatment naïve provided 48 samples that were profiled with scRNA+TCRαβ-seq with from CD45+ enriched tumor biopsies, and the data was acquired from GEO with accession number GEO: GSE120575.

From the Riaz et al. data[6], 65 patients with metastatic melanoma who were treated with anti-PD1 provided 109 prior and post-therapy samples that were profiled with bulk-RNA-seq with from tumor biopsies, and the data was acquired from https://github.com/riazn/bms038_analysis.

### TCRβ-seq data analysis

Analyses started with the TCRβ matrices provided by the Adaptive Biotechnologies preprocessing pipeline. Matrices were quality controlled with VDJtools[53] (ver 1.2.1), where non-functional clonotypes were removed and diversity indices calculated with CalcDiversityStats-function. Multiple different diversity metrics, including Shannon-Wiener, Simpson, and clonality indexes were calculated with CalcDiversityStats-function on both unsampled and subsampled repertoire data. The same TCRs across samples were pooled together with JoinSamples-command.

The expanded clonotypes were defined with Fisher's two-sided exact test, where the unnormalized read-depths from pre and post therapy samples were used an input, and the P-values were corrected with Benjamini-Hochberg adjustment. An expanded clonotype was defined as $P_{adj} < 0.05$.

The clonally replaced clones were determined as clones that were found only in the post-therapy samples, and thus the frequency of clonally replaced clones was calculated as: 1 − frequency of persisting

clones present both in the pre and post samples, and their clonality as in the rest of the repertoire level samples.

## Unsupervised determination of epitope-specificities of TCRs with GLIPH 2.0

TCRs were grouped based on amino acid level -similarities decided by GLIPH2[25] (ver 1.0.0) with default parameters and CD8 as reference sets for CD8+-sorted samples and CD4CD8 for MNC-sorted samples.

## Supervised determination of epitope-specificities of TCRs with TCRGP

Epitope-specificity predictions were performed with TCRGP (ver 1.0.0). To build TCRGP models specific for MAA epitopes (MART1$_{AAGIGILTV}$, MART1$_{ELAGIGILTV}$, and MELOE1$_{TLNDECWPA}$), we pooled all available anti-MAA TCRα or TCRβ sequences together, respectively, specific to that epitope from individual donors with a randomly selected set of background TCRs, so that there was always an equal number of epitope-specific and background TCRs. To assess the models, we used either leave-one-out cross-validation with epitopes with <100 epitope-specific TCRs and 20-fold cross-validation for epitopes with more TCRs (Supplementary Data 1). The anti-MAA classifiers were also assessed in leave-one-subject-out cross-validation scheme (Supplementary Fig. 5b) and as precision-recall AUROCs with 1:1 of positive and negative samples, from which we have also included the Area under the Precision-Recall Curves (AP-values[54]) as summary values (Supplementary Fig. 6a). All other models were gathered from the TCRGP packages GitHub page for each TCRβ identified in the dataset. The tested epitopes were "GILGFVFTL_cdr3b" (from Influenza A M1 $_{GILGFVFTL}$ epitope), "GLCTLVAML_cdr3b" (EBV BMLF1$_{GLCTLVAML}$ epitope), "IPSINVH-HY_cdr3b" (CMV pp65$_{IPSINVHHY}$ epitope), "NLVPMVATV_cdr3b" (CMV pp65$_{NLVPMVATV}$ epitope), "RAKFKQLL_cdr3b" (EBV BZLF1$_{RAKFKQLL}$ epitope), "RPRGEVRFL_cdr3b" (HSV2 B7$_{RPRGEVRFL}$ epitope), "TPRVTGGGAM_cdr3b" (CMV pp65$_{TPRVTGGGAM}$ antigen), and "YVLDH-LIVV_cdr3b" (EBV BRLF1$_{YVLDHLIVV}$ epitope).

For the predictions used in all analyses, a threshold corresponding to a false positive rate (FPR) of 5% was determined for each epitope separately from the ROC curves obtained from the cross-validation experiments. To determine anti-MAA T cells, we have predicted each of the 5 classes individually, and if any of these classes are positive, the T cell is labeled as anti-MAA. The FPR was tuned to be 5% for individual epitopes, and thus the FPR for anti-MAA T cells is larger than 5%. The overall FPR to determine anti-MAA T cells was calculated by predicting the 59 898 human non-anti-MAA TCRs found in the VDJdb[26] in January 2021 and the TCR was considered as a false-positive if the TCR was predicted to recognize any of the 5 MAA epitopes, and the value was 0.234, suggesting that our 5 TCRGP classifiers have a limited correlation for detecting the same TCRs as negative.

## scRNA + TCR(α)β-seq data analysis

**Metastatic melanoma biopsies from untreated patients (Li et al. data).** The data contains tumor-infiltrating immune cells and residual tumor cells from melanoma biopsies from treatment naïve or immunotherapy resistant patients. The quality control was used from the original publication. The analysis started with a count matrix, which was log-normalized with a scaling factor of 10,000, and the top 2000 genes with the highest variability were determined with FindVariableFeatures-command implemented in Seurat[55] with method "vst". From the highly variable genes, genes that were related to V(D)J-recombination and mitochondrial transcripts were excluded, and the remaining genes were fed into principal component analysis (PCA), where the components with standard deviation above 2 were retained and used for non-linear uniform manifold approximation and projection (UMAP) dimensionality reduction[56] with RunUMAP-function implemented in Seurat with default parameters and to graph-based clustering implemented in Seurat with default parameters. Based on

visual analysis, the resolution of clustering was chosen as 0.3 to avoid over- and under-clustering of the dataset. The clusters were named in descending order (cluster 0 contains the most cells) annotated based on analysis of cluster adjacency, DE-genes, canonical markers, expression of TCR, reference-based cell-type annotation with SingleR[57] (ver 1.2.4) with Blueprint as a reference, and with the original cell phenotype annotation used in the original publication.

**Metastatic melanoma biopsies from untreated or treated patients (Jerby-Arnon et al. data).** The data contains tumor-infiltrating immune cells and tumor cells from melanoma biopsies prior and post-treatment. TILs and tumor cells were analyzed separately. The quality control was received from the original publication and the data was processed similarly as with Li et al. data, albeit no additional clustering was performed.

**Uveal metastatic melanoma biopsies from untreated patients (Durante et al. data).** The data contains tumor-infiltrating immune cells and tumor cells from uveal melanoma biopsies from one time point. Cells with >15% mitochondrial transcripts, <100 or >8500 expressed genes, or <400 UMI counts were removed from the analysis. To overcome batch-effect, we used scVI[58] (ver 0.5.0) with default parameters ($n$_hidden=128, $n$_latent=30, $n$_layers=2, dispersion = 'gene') where each sample was treated as a batch. The obtained latent embeddings were then used for graph-based clustering and UMAP visualization as with the Li et al. data, and the resolution was chosen as 0.5. Data was scaled and normalized as with the Li et al. data, and clusters were annotated with the same approach.

**Metastatic melanoma biopsies before and after immune checkpoint blockade (anti-PD1 or anti-PD1 + anti-CTLA4) (Sade-Feldman et al. data).** The data contains tumor-infiltrating immune cells and from patients with melanoma from melanoma biopsies prior and post-treatment. The quality control was received from the original publication and the data was processed as with Li et al. data, albeit no additional clustering was performed.

**General.** Differential expression analyses were performed based on the t-test, as suggested by Soneson et al.[59]. Pathway analyses were done with either hypergeometric test on GO- and HALLMARK-categories gathered from MSigDB with Gene Set Enrichment Analysis (GSEA) with R-package clusterProfiler[60] (ver 3.16.0). Receptor–ligand interactions were calculated with CellPhoneDB[33] (ver 2.0.0) with at least 50 cells and 1000 iterations for the permutation testing, where the cell numbers were normalized across phenotypes. The costimulatory and coinhibitory receptor-ligand pairs were gathered from Dufva and Pölönen et al.[61].

## Survival analysis

Survival analysis was conducted with the Kaplan–Meier method, and continuous data were split at the median. The log-rank test was used to determine statistical significance. The multivariable analysis was performed with Cox proportional hazards model with R-package finalfit (v 1.0.5), where the overall survival was used as the dependent variable and the presence of MART1-specific dominating clone (dichotomous), presence of any dominating clone (dichotomous), sex (dichotomous), melanoma type (factor), TNM class (factor) Breslow thickness (continuous), age (continuous), mitotic rate (continuous), and clonality (continuous) were used as explanatory variables. All the explanatory variables were also analyzed in a univariate setting.

## Statistical testing

P-values were calculated with nonparametric tests, including Mann–Whitney test (two groups), Kruskal–Wallis test (more than two groups), and Fisher's exact test where the alternative hypotheses are reported. P-values were corrected with Bonferroni (differentially

expressed genes) or with Benjamini–Hochberg (all other tests) adjustment. All calculations were done with R (4.0.2) or Python (3.7.4).

### Data visualization
In the box plots, the center line corresponds to the median, box corresponds to the interquartile range (IQR), and whiskers 1.5 × IQR, while outlier points are plotted individually where present.

### Reporting summary
Further information on research design is available in the Nature Research Reporting Summary linked to this article.

## Data availability
The TCRβ-sequencing data and Seurat-objects are available at Zenodo under DOI: 10.5281/zenodo.6882576 with restricted access due to General Data Protection Regulation (GDPR) regulations and data can be accessed by placing a request via Zenodo to the leading and corresponding authors and will be reviewed without undue delay. Additionally, the TCRB-sequencing data are available at immuneAccess under [https://doi.org/10.21417/JH2022NC]. The publicly available scRNA+TCRαβ-sequencing and TCRβ-sequencing data used in this study are listed in Supplementary Data 1. Source data are provided with this paper. The remaining data are available within the Article, Supplementary Information or Source Data file. Source data are provided with this paper.

## Code availability
The code to reproduce the key findings is available in [https://github.com/janihuuh/melanomap_manu]; v1: https://doi.org/10.5281/zenodo.6875637

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

## Acknowledgements

We are deeply grateful to all patients who participated in the study and generously contributed samples. We acknowledge the computational resources provided by the Aalto Science-IT project and all the personnel in the Hematology Research Unit Helsinki for insightful conversations. We especially thank Dr. Nathalie Labarriere for providing information of MAA-specific TCR sequences of their study; Dr. Michael Durante and Dr. J. William Harbor for providing information related to their study; Dr. Olli Dufva for his help with interpreting the data. We additionally acknowledge Ms. Lotta Länsman for providing administrative advice. This study was supported by the European Research Council (Project: M-IMM 647355)(SM), Academy of Finland (314442, S.M., H.L. and 335527 S.M.), Sigrid Juselius Foundation (S.M.), Signe and Ane Gyllenberg Foundation (S.M.), Helsinki Institute for Life Science (S.M.), Cancer Foundation Finland (S.M.) and State funding for the University-level Health Research in Finland (S.M.). J.H. was supported by Finnish Hematology Association, Blood Disease Research Foundation, Helsinki Institute for Life Science, Biomedicum Helsinki Foundation, Finnish Medical Foundation, K. Albin Johansson Foundation, Kaute Foundation, Suomalais-Norjalainen Lääketieteen Säätiö, and Emil Aaltonen Foundation. T.L. was supported by Academy of Finland (Decision 311081).

## Author contributions

The study was conceived by J.H., L.C., M.M.D., and S.M. The MART1-specific TCRβ-seq data were collected by C.Y. The new samples from melanoma patients used in TCRβ-seq data (Helsinki cohort) were collected and processed by J.H., T.L., H.K., A.K., M.H., and K.P. Rest of the data was collected by J.H. from public repositories. All the analyses were designed, performed, and interpreted by J.H. with help from L.C., E.J., C.W., H.L., M.M.D., and S.M. The figures were drafted by J.H. with contributions from the rest of the authors. The manuscript was written by J.H., with the help of S.M., M.M.D., and H.L., and with contributions from other authors. The project was supervised by H.L., M.M.D., and S.M.

## Competing interests

M.H. has received honoraria from Pierre Fabre, Novartis, Bristol-Myers Squibb (B.M.S.), Merck Sharp & Dohme (M.S.D.), Roche, Amgen, Sanofi, and Incyte. S.M. has received honoraria and research funding from B.M.S. and research funding from Novartis and Pfizer. M.M.D. has received honoraria and/or equity from Amgen, Chugai, Vir, IGMS, Janux, A2, 3T, Mozart, Red Tree, and research funding from Sanofi. The remaining authors declare no competing interests.
