## [Peer Review File · Nature Communications]

Evolution and modulation of antigen-specific T cell responses in melanoma patientsEditorial Note: This manuscript has been previously reviewed at another journal that is not operating a transparent peer review scheme. This document only contains reviewer comments and rebuttal letters for versions considered at Nature Communications.

REVIEWERS' COMMENTS

Reviewer #3 (Remarks to the Author):

The original comments raised by myself (reviewer 3) in the previous two rounds of review have now been addressed, particularly the question of age differences in cases vs controls. The updated age matched subsampling analysis shows a clear difference.

No further points are raised from me, and I echo the previously stated positives on the manuscript (reproducibility: the manuscript code was available for full review, novel analysis: larger sample sizes and more in-depth analysis than prior work in this area).

Reviewer #3 (Remarks to the Author):

The original comments raised by myself (reviewer 3) in the previous two rounds of review have now been addressed, particularly the question of age differences in cases vs controls. The updated age matched subsampling analysis shows a clear difference.

No further points are raised from me, and I echo the previously stated positives on the manuscript (reproducibility: the manuscript code was available for full review, novel analysis: larger sample sizes and more in-depth analysis than prior work in this area).

We thank the reviewer for their thorough evaluation of our manuscript during different stages and aiding us in making it better.